# SPIKE-TO-EXCITE: PHOTOSENSITIVE SEIZURES IN BIOLOGICALLY-REALISTIC SPIKING NEURAL NETWORKS

## ABSTRACT

Photosensitive Epilepsy (PE) is a neurological disorder characterized by seizures triggered by harmful visual stimuli, such as flashing lights and high-contrast patterns. The mechanisms underlying PE remain poorly understood, and to date, no computational model has captured the phenomena associated with this condition. Biologically detailed spiking networks trained for efficient prediction of natural scenes have been shown to capture V1-like characteristics. Here, we show that these models display seizure-like activity in response to harmful stimuli while retaining healthy responses to non-provocative stimuli when post-synaptic inhibitory connections are weakened. Notably, our adapted model resembles the motion tuning and contrast gain responses of excitatory V1 neurons in mice with optogenetically reduced inhibitory activity. We offer testable predictions underlying the pathophysiology of PE by exploring how reduced inhibition leads to seizure-like activity. Finally, we show that artificially injecting pulsating input current into the model units prevents seizure-like activity and restores baseline function. In summary, we present a model of PE that offers new insights to understand and treat this condition.

## 1 INTRODUCTION

In 1997, a *Pokemon* episode featuring rapid explosion scenes triggered seizures in over 600 children with Photosensitive Epilepsy (PE) (Hermes et al., 2017). PE is characterised by abnormal sensory processing in the primary visual cortex (V1) in response to specific visual stimuli, such as flickering lights and high-contrast features (Fisher et al., 2022; Seshia & Carmant, 2005) (Fig. 1a). This results in seizure-like activity, defined as excessive neuronal firing (Fig. 1b). Extensive research suggests that one of the main causes underlying seizures is weakened GABA-mediated transmission at inhibitory synapses (Staley, 2015; Prince & Wilder, 1967; McCormick & Contreras, 2001) (Fig. 1c), as supported by the efficacy of numerous anti-epileptic drugs that enhance GABAergic function (Shao et al., 2019). However, the exact mechanisms underlying PE remain unclear, partly due to the lack of an accurate animal or computational model that accurately captures the human condition (Hermes et al., 2017; Cossart et al., 2005).

Most existing computational work in epilepsy has focused on identifying seizure-like activity (Kuhlmann et al., 2018) and seizure onset zones for surgical resection (Saggio & Jirsa, 2024). The few investigations that model seizure-like activity have fixated on emulating general seizure dynamics in non-visual areas like the hippocampus and widespread cortical areas (Jirsa et al., 2014; Liou et al., 2020) instead of capturing seizures induced by photic stimulation in the visual cortex. There are several models of seizure dynamics using dynamic systems theory (Da Silva et al., 2003; Du et al., 2019). However, these models do not emulate underlying biophysical processes, making it difficult to link abstract model variables to biological mechanisms (Liou et al., 2020). An alternative model (Liou et al., 2020) was proposed which does incorporate key biophysical elements (such as recurrent projections), however, it lacks other biological characteristics, such as spontaneous seizure initiation, relying instead on focal excitatory inputs to initiate seizure-like activity. To date, no biologically realistic model captures the response properties of PE.

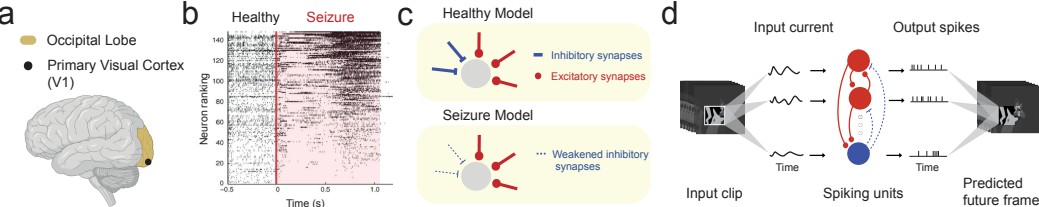

Figure 1: **The photosensitive seizure model. a.** Schematic of the primary visual cortex (V1) in the occipital lobe, the brain region modelled in this study, and the seizure onset zone in PE. **b.** Neuronal spike raster of patient recordings from the occipital cortex, adapted from (Truccolo et al., 2011). **c.** Illustration comparing synaptic connections in the control model to the weakened synaptic connections in the seizure model. **d.** Schematic of the V1 seizure model adapted from (Taylor et al., 2024b). The model converts a sequence of movie frames into an input current, fed into a single hidden layer of recurrently connected spiking Leaky Integrate-and-Fire (LIF) units with weakened inhibitory connections. The spike output linearly constructs a prediction of the input stimulus of the future.

A recent modelling study (Taylor et al., 2024b) developed a recurrently connected spiking network of excitatory and inhibitory units, trained for efficient temporal prediction of natural movie clips (Fig. 1d). This model captures simple and complex cell-like tuning characteristics and highlights differences between excitatory and inhibitory V1 neurons. Given the model's resemblance to V1, and the weakened inhibitory transmission theory underlying epilepsy, we aimed to explore the following question: **Do weakened inhibitory connections in the V1 spiking temporal-prediction model result in response characteristics similar to those found in photosensitive epilepsy?** Our contributions are summarized as follows:

- To the best of our knowledge, we establish the first computational model of PE. We demonstrate how the spiking temporal-prediction model (Taylor et al., 2024b), with weakened inhibitory synapses, captures seizure-like activity in response to known seizure-inducing stimuli, while retaining normal V1-like activity to non-provocative stimuli.

- We found our seizure model to capture V1-like spike statistics during non-seizure periods, with the model's tuning properties and contrast responses resembling those of V1 neurons with weakened inhibitory input.

- We investigated mechanisms that trigger and sustain seizure-like activity in our model, finding the weakened inhibitory synapses to increase response gain. As a result, the model was more susceptible to abnormally high firing rates in response to high-contrast stimuli, with seizure-like activity sustained by an imbalance in excitatory and inhibitory currents.

- Notably, we managed to prevent seizure-like activity by inputting pulsating currents into model units, akin to the responsive neurostimulation (RNS) technique used in the treatment of epilepsy (Krauss et al., 2021). Our findings may be clinically relevant to optimizing neural stimulation protocols, and be a suitable *in silico* candidate for further exploring seizure mechanisms and clinical therapies.

## 2 METHODS

**The V1 spiking temporal-prediction model with weakened inhibitory synapses** We implemented the spiking V1 seizure model using the pre-trained spiking V1 temporal-prediction model (Taylor et al., 2024b) with weakened inhibitory connections to reflect hypothesized GABAergic abnormalities in PE networks (see Appendix 5.1 for details). We used this model as it incorporates important biological properties (spikes and Dale's law); is trained using a biologically plausible (Keller & Mrsic-Flogel, 2018) and unsupervised normative objective of efficient future-frame prediction of natural movies (as also explored in other works (Singer et al., 2023; Taylor et al., 2024c)); and captures V1-like spike statistics, motion tuning and various key differences between excitatory and inhibitory neurons. In comparison, other models omit spikes (Echeveste et al., 2020); hard-code

connectivity (Hennequin et al., 2018); operate in the spatial domain (Zylberberg & DeWeese, 2013); or explore predictive objectives in non-visual brain regions like the hippocampus (Levenstein et al., 2024). We report all results using the model with $80\%$ weakened inhibitory connections. However, we found a range (50-100%) of inhibitory weight decreases to result in seizure-like activity in response to provocative stimuli (Appendix Fig. 1).

**Datasets** We used three different visual datasets for model validation. 1. We collected a dataset of various $\sim 10$ second long movie scenes (*Pokemon Shock* clip, Kanye West's *All of the Lights* music video, a *Citroen* advert, the *Incredibles 2* movie and The Weeknd's *Take My Breath* music video) which have been reported to induce seizures in patients with PE ((Concepcion, 2013; Sweney, 2012; Svachula, 2018; Aswad, 2021)) to assess if our model also exhibits seizure-like activity; 2. We used segments of the *Star Wars* movies (8000 patches of $\sim 2$ second duration) for comparing model spike responses to previously published monkey V1 data (Rasch et al., 2011); 3. We used a natural stimulus dataset (2000 patches of $\sim 10$ second duration) (Taylor, 2023) containing various natural scenes, which we used to analyse model contrast tuning and excitatory-inhibitory (EI) balance. To match the model's training data (Taylor et al., 2024b), all validation stimuli were grayscaled; sampled at (or sampled to) 120Hz, and bandpass filtered to emulate the retinal and thalamic transformations; normalized; and clamped to range $[-3.5, 3.5]$.

**Seizure identification in the model and classifying stimuli** Seizures are usually identified through clinical observation (*e.g.* loss of consciousness or jerking movements). Patients with photosensitive epilepsy (PE) are diagnosed with electroencephalogram (EEG) to identify abnormally elevated activity in response to photic stimuli (Fisher et al., 2005). These diagnostic measures are not directly possible in our model and hence we used the model's spiking activity to identify seizure-like activity. Prior works define seizure activity as an abnormally large increase in population activity (Schevon et al., 2012). We confirmed if the model's activity was abnormally high on our collected dataset of known provocative movie stimuli (Appendix Fig. 1) and was not elevated to non-provocative stimuli. We found a small portion of clips to evoke abnormally high population activity to the natural stimulus dataset in the seizure model (and not in the control model) and we classified these clips as provocative if the model's mean population firing rate surpassed a threshold of 84Hz (Appendix Fig. 2). Note, due to ethical reasons we could not confirm if these clips induced seizures in PE patients. As additional comparisons, we calculated spectrograms from the local field potential (LFP) approximations of the model's mean population spiking responses (Barbieri et al., 2014; Mazzoni et al., 2008), which represent the same biophysical process as EEG recordings (Buzsáki et al., 2012). Furthermore, we also calculated the synchronicity of the model's spiking activity, where increased neural synchrony has been noted as a characteristic of seizures (Schevon et al., 2012) (although this remains controversial (Truccolo et al., 2011)).

**Spike statistics and motion tuning** To be comparable to (Rasch et al., 2011), we calculated all spike statistics using a time window of 2s with a step size of 0.2s in response to the *Star Wars* movies. As done in (Taylor et al., 2024b), we quantified the correlation between different units' activity by calculating the average correlation coefficient between the spike trains (binned at 25ms) of 400 randomly sampled unit pairs for varying temporal lag. We calculated the orientation selectivity index $\text{OSI} = (R_{\text{pref}}^{\text{orient}} - R_{\text{orth}}^{\text{orient}})/(R_{\text{pref}}^{\text{orient}} + R_{\text{orth}}^{\text{orient}})$ and the direction selectivity index $\text{DSI} = (R_{\text{pref}}^{\text{dir}} - R_{\text{non-pref}}^{\text{dir}})/(R_{\text{pref}}^{\text{dir}} + R_{\text{non-pref}}^{\text{dir}})$ (Niell & Stryker, 2008; Rochefort et al., 2011), where $R_{\text{pref}}^{\text{orient}} = R_{\text{pref}}^{\text{dir}}$ is a unit's mean response to a stimulus in its preferred orientation and direction, and $R_{\text{orth}}^{\text{orient}}$ and $R_{\text{non-pref}}^{\text{dir}}$ are the mean responses in the orthogonal orientation and opposite direction to the preferred direction, respectively. Both measures range between zero and one, where one corresponds to maximum selectivity. We generated various drifting full-field sinusoidal gratings to assess tuning responses (as implemented in (Taylor et al., 2024b)).

**Contrast tuning and receptive fields** We increased stimulus contrast by multiplying each pixel by scalar $c$ and clamping all values to range $[-3.5, 3.5]$ (the pixel range of the model's training data). We used scalar $c \in [1, 2, 3, 4]$ to increase contrast in the natural stimulus dataset and used $c$ logarithmically spaced between $0.75$ and $4.0$ for the unit contrast tuning analysis (to be more comparable to the work of (Atallah et al., 2012)). Model receptive fields (RFs) were estimated using a spike-triggered average to white-noise clips (Dayan & Abbott, 2005).

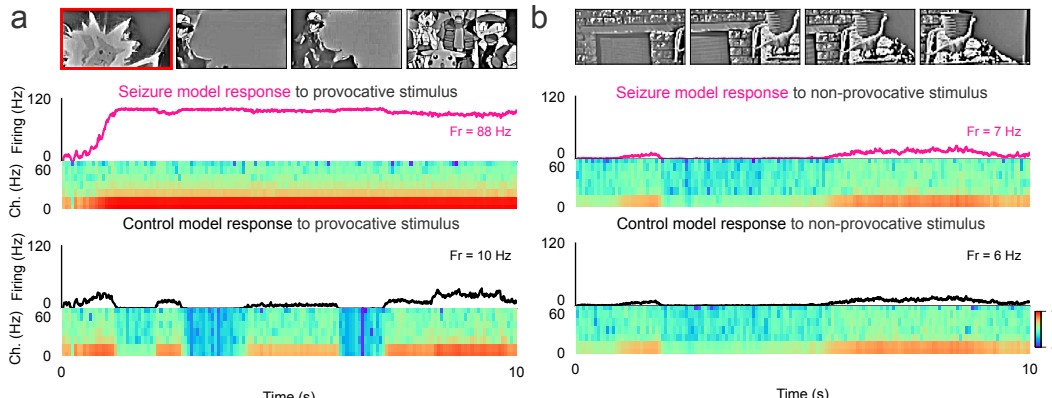

Figure 2: **Capturing seizure-like activity to provocative stimuli and normal V1-like activity to non-provocative visual input. a.** Mean population firing activity of the seizure model (pink) and control model (black) in response to the *Pokemon Shock* episode (provocative frame highlighted in red). Spectrograms plot the frequency-time power representation of the model activity, with the highest and lowest power denoted in red and blue respectively. **b.** Corresponding results in response to non-provocative natural stimuli.

**Excitatory and inhibitory currents** We quantified the global (*i.e.* how equal) and precise (*i.e.* how coupled) EI balance of the model units (Vogels et al., 2011; Taylor et al., 2024b). A unit's global EI balance was measured as the ratio $\sum_t \text{Ex}[t]/|\text{In}[t]|$ between its excitatory (Ex) and inhibitory (In) currents over time; and a unit's precise EI balance was quantified as the correlation between its (negative) inhibitory and (positive) excitatory currents. Before calculating the precise EI balance, we convolved each current time series using a Gaussian kernel ($\sigma = 72$ms) (Dayan & Abbott, 2005). We smoothed the current time series to emulate synaptic conductance (Gerstner et al., 2014), as done in (Taylor et al., 2024b) (see Appendix Fig. 4 for EI correlations calculated using different Gaussian kernel $\sigma$).

**RNS-like stimulation in the model** We replicated neural stimulation techniques for treating drug-resistant epilepsy by inputting pulsating current into the model units. Current stimulation in our model resembles RNS, where electrodes are implanted near the epileptic focus in the cortex or hippocampus (Krauss et al., 2021), unlike deep brain stimulation, which targets the thalamus (Foutz & Wong, 2022). RNS stimulation parameters range between 1-3mA with a pulsation frequency up to 200Hz (Bigelow & Kouzani, 2019). We explored the space of stimulation frequencies logarithmically spaced between 0-120Hz and amplitudes logarithmically spaced between 0-1.28a.u. to determine the optimal values for reducing seizure-like activity in response to the natural stimulus dataset.

## 3 RESULTS

### 3.1 CAPTURING SEIZURE-LIKE ACTIVITY TO PROVOCATIVE STIMULI

We examined the control and seizure model's firing rate to different stimulus inputs and found their responses to qualitatively differ from provocative stimuli, whilst appearing similar to non-provocative stimuli. Here, we used a known seizure-inducing movie clip as provocative stimuli and a natural movie recording containing no high-contrast and flashing visual features as (presumed) non-provocative stimuli. We report additional results to other known seizure-inducing movie clips in Appendix Fig. 3, including Kanye West's *All of The Lights* music video, a *Citroen* advert, Disney's *Incredibles 2* movie and The Weeknd's *Take My Breath* music video - all reported to induce seizures in viewers (Concepcion, 2013; Sweney, 2012; Svachula, 2018; Aswad, 2021).

The seizure model exhibited a high (88Hz) and non-decaying population firing rate in response to the infamous seizure-inducing *Pokemon* scene from 1997 (Hermes et al., 2017). In contrast, the control model displayed a lower (10Hz) and temporally-varying population firing rate, resembling

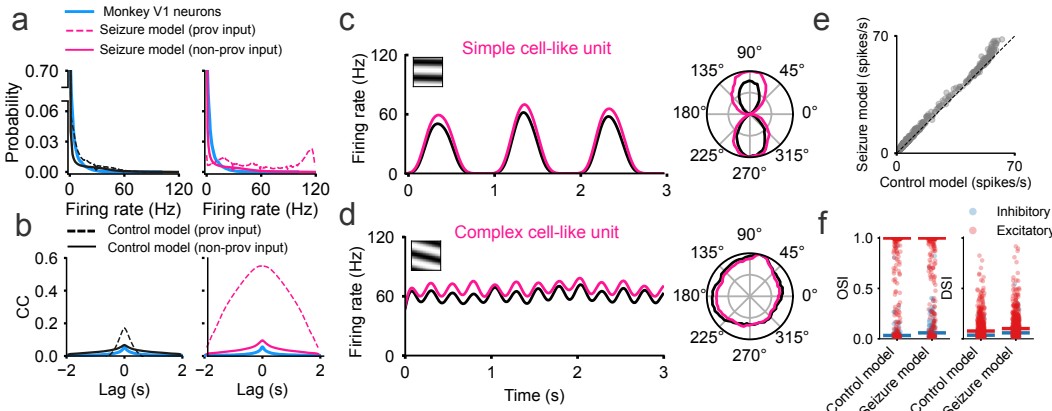

Figure 3: **V1-like spike statistics and tuning characteristics in the seizure model. a.** Firing rate distribution and **b.** cross-correlograms (measuring the correlation coefficient between the activity of randomly sampled units) for provocative (dashed black) and non-provocative stimuli (black) in the control model, and provocative (dashed pink) and non-provocative (pink) stimuli in the seizure model, and experimental data from monkey V1 neurons (blue) (Rasch et al., 2011). **c.** Simple cell-like and **d.** complex cell-like firing patterns in response to optimal gratings (inset) in the seizure (pink) and control model (black). **e.** Mean firing rate of every unit in response to its optimal grating in the seizure (y-axis) and control model (x-axis). **f.** Orientation selectivity index (OSI) and direction selectivity index (DSI) of the excitatory (red) and inhibitory (blue) units in the seizure and control model, where bars represent median selectivity.

normal V1 activity to natural stimuli (Rasch et al., 2011) (Fig. 2a). However, both models responded similarly to the non-provocative stimulus clip, with a population firing rate of $\sim 6 - 7$Hz (Fig. 2b).

EEG recordings are one technique routinely used to identify seizures in patients with epilepsy (Fisher et al., 2005). We approximated these recordings using spectrogram representations of the model's spiking activity (see Methods). The seizure model's spectrogram differed from the control model in response to provocative stimulus, with a higher power in the frequency range $0 - 20$Hz (Fig.2a). This qualitatively resembles the spectrogram properties of LFP recordings from occipital and middle temporal cortices during seizures in humans with focal epilepsy (Truccolo et al., 2011). There was no difference between the seizure and control model spectrograms for the non-provocative stimulus (Fig.2b).

### 3.2 EMERGENCE OF V1-LIKE SPIKE STATISTICS AND TUNING PROPERTIES

We examined the seizure model's spike statistics and tuning properties, and found several V1-like characteristics to emerge. We analyzed the model's spike activity in response to *Star Wars* movie clips (Rasch et al., 2011), classifying clips as provocative if they triggered seizure-like activity, or non-provocative, if they did not (see Methods). The seizure model displayed a firing rate distribution with an exponential fall-off (Fig. 3a) to the non-provocative stimuli as similarly found in the control model (Taylor et al., 2024b) and V1 data (Rasch et al., 2011; Baddeley et al., 1997). Quantifying the correlation statistics using spike train cross-correlograms (Rasch et al., 2011), revealed the correlation between different unit spike trains to exhibit a low peak correlation that symmetrically decays for increasing lag (Fig. 3b). This resembles the correlation statistics of the control model and of anesthetized monkey V1 (Rasch et al., 2011). However, the seizure model's spike statistics substantially changed in response to the provocative stimuli. The firing rate distribution peaked at $\sim 100$Hz (Fig. 3a) and the peak correlation increased from $0.04$ to $0.55$ CC (Fig. 3b), reminiscent of seizure-like activity (Schevon et al., 2012). In comparison, the control model's spike statistics remained largely unaltered.

We assessed unit tuning statistics using responses from full-field drifting sinusoidal gratings. As reported in the control model (Taylor et al., 2024b), the seizure model units also exhibited response properties resembling the two prominent cell classes in V1, namely simple (Fig. 3c) and complex

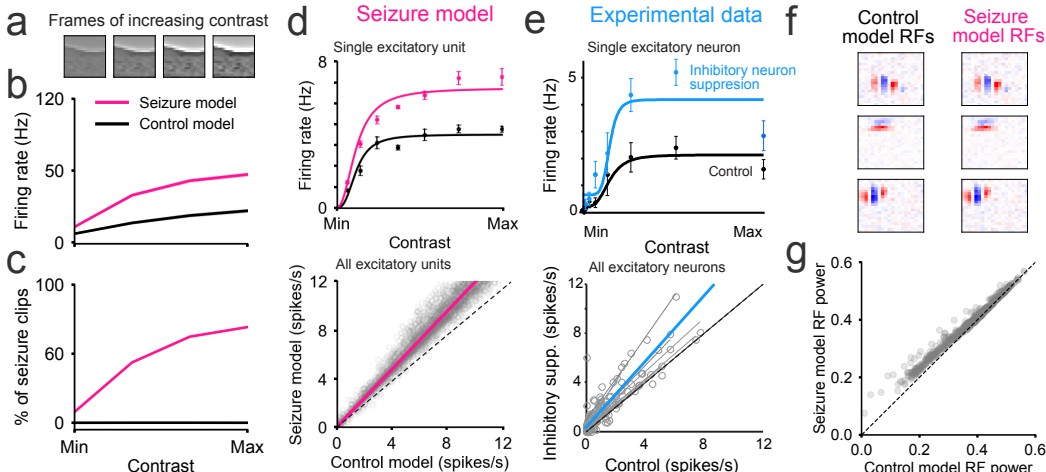

Figure 4: **Seizure onset induced by high-contrast stimuli. a.** Example stimulus frames of increasing contrast. **b.** Change in firing rate and **c.** percentage of clips inducing seizure-like activity in the seizure (pink) and control (black) models for increasing stimulus contrast. **d.** Top: contrast tuning curve of a single unit of the seizure (pink) and control model (black). Error bars are the SEM. Bottom: Firing rates of the excitatory seizure model (y-axis) versus the control model (x-axis). Circles plot individual unit responses at different contrasts. Gray lines plot the linear fit for each unit and the pink line plots the average linear fit. **e.** Same as in **d.**, but for excitatory V1 neuron responses (black) plotted against excitatory V1 neuron responses with inhibitory neurons optogenetically suppressed (blue) (Atallah et al., 2012). **f.** Example spatial receptive fields (RFs) from the control and seizure models (red = excitatory and blue = inhibitory). **g.** RF power (largest mean squared values of a spatial RF in time) of the seizure (y-axis) and control model (x-axis) units.

cells (Fig. 3d). Both cell responses oscillate in response to their preferred grating, with more pronounced response fluctuations in simple cells (Movshon et al., 1978a) compared to complex cells, whose responses are less influenced by grating phase (Movshon et al., 1978b). In comparison to the control model, we observed the seizure model's responses to be more elevated, with the units having a higher firing rate to their preferred grating (Fig. 3e).

The orientation and direction selectivity remained similar between the seizure and control model (Fig. 3f; albeit statstically different, see Appendix Table 1 for p-values). Orientation and direction selectivity were quantified using the orientation selectivity index (OSI) and direction selectivity index (DSI), respectively (Niell & Stryker, 2008; Rochefort et al., 2011) (see Methods). The inhibitory seizure model's units were more broadly tuned than the excitatory units and both neuron classes displayed little direction selectivity, as has similarly been reported in mouse V1 (Niell & Stryker, 2008).

The seizure model's increased response amplitude and modestly altered motion tuning properties mirror biological observations (Atallah et al., 2012; Katzner et al., 2011). These phenomena emerge in the seizure model due to weakened inhibitory connections, which reduce the inhibitory population's influence on their excitatory counterpart. Similarly, V1 response gain increases with minor variation in tuning properties when inhibitory neuronal activity is optogenetically (Atallah et al., 2012) or pharmacologically (Katzner et al., 2011) reduced, as reported in mouse and cat V1, respectively.

### 3.3 HIGH-CONTRAST STIMULI TRIGGER SEIZURE ONSET

High-contrast stimuli are believed to trigger photic seizures (Hermes et al., 2017). To test if this is also the case in our model, we increased the contrast levels of the natural stimulus dataset (Fig. 4a) (see Methods) and quantified the percentage of clips that resulted in seizure-like activity. Although the firing rates increased in both the control and seizure models for increasing stimulus contrast (Fig. 4b), only the seizure model exhibited seizure-like activity (Fig. 4c). The occurrence of seizure-

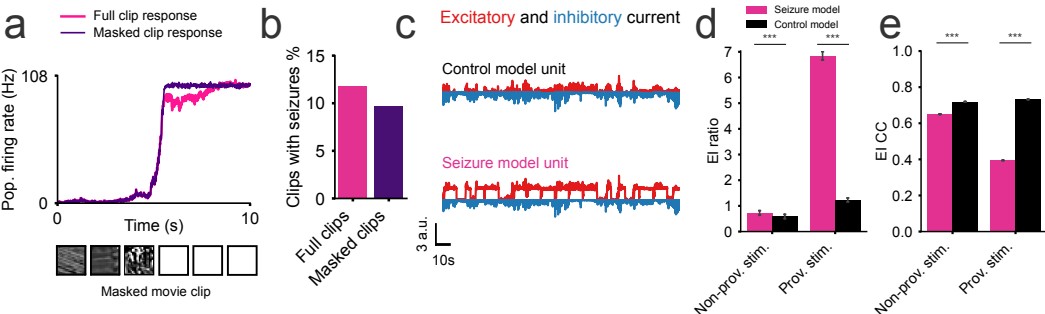

Figure 5: **Seizure propagation sustained by recurrent dynamics with an imbalance in excitatory and inhibitory currents. a.** Population firing rates of the seizure model to full clip (pink) and masked clip (purple), with input frames masked to zero after seizure onset. **b.** Percentage of clips where seizure-like activity persists after masking the input clips. **c.** Excitatory (red) and inhibitory (blue) currents to a unit in the control (top) and seizure model (bottom). **d.** EI ratio in the control and seizure model in response to the provocative and non-provocative stimuli. **e.** Corresponding EI correlation coefficients. Bars in **d.** and **e.** plot the median and bootstrapped standard error, with statistical significance assessed using the Mann-Whitney U test (***p< 0.001).

like activity grew with increasing stimulus contrast, aligning with the observations of high-contrast stimuli triggering seizure onset (Hermes et al., 2017).

V1 neurons exhibit a growing response curve shaped like the hyperbolic ratio equation (Albrecht & Hamilton, 1982), in response to visual gratings of increasing contrast. Excitatory V1 neuron contrast-response curves are reported to rise in amplitude as the influence of V1 inhibitory neurons is optogenetically (Atallah et al., 2012) and pharmacologically (Katzner et al., 2011) reduced. The contrast-response curves of the control and seizure model units appeared to be well-fitted by the hyperbolic ratio equation on visual inspection (Fig. 4d). Consistent with biology, the seizure model with weakened inhibitory synapses exhibited higher contrast responses than the control model ($p = 8.28 \times 10^{-126}$, t-test), across the population of excitatory units (Fig. 4d and e).

To understand how weakened inhibitory synapses in the seizure model led to increased responsiveness to high-contrast visual input, we inspected each model's receptive fields (RFs). The seizure model's RF shapes were unaltered compared to the control model ($CC = 0.99$), and resembled the RFs of V1 simple cells (Hubel & Wiesel, 1959; 1968) (Fig. 4f), with varying numbers of excitatory and inhibitory subfields tuned for a particular spatial frequency and orientation (Jones & Palmer, 1987; Ringach, 2002). However, the seizure model's RF exhibited higher power compared to the control model (Fig 4g), where RF power quantifies the response sensitivity of a unit's RF to visual input (Taylor et al., 2024b). This indicates that the weakened inhibitory synapses in the seizure model enhance the response gain of the units, rendering the model more susceptible to heightened activity.

### 3.4 EXCITATION-INHIBITION IMBALANCE SUSTAIN SEIZURE-LIKE ACTIVITY

We explored whether the seizure model's prolonged seizure-like activity was primarily sustained by the provocative input stimulus, or rather by an imbalance in recurrent dynamics. We investigated this by masking the provocative input stimuli following the onset of seizure-like activity (Fig. 5a). The fraction of provocative clips only marginally reduced from $11.8\%$ to $9.7\%$ (Fig. 5b), implying the seizure-like activity to be driven by an imbalance in recurrent dynamics.

The balance of excitatory and inhibitory (EI) synaptic currents received by cortical neurons (Okun & Lampl, 2008; Wehr & Zador, 2003; Anderson et al., 2000; Monier et al., 2003; Xue et al., 2014) is thought to be critical for maintaining stable neural activity (Sadeh & Clopath, 2021). We qualitatively observed the excitatory input currents to be more elevated in the seizure model compared to the control model in response to provocative stimuli (Fig. 5c). The EI balance can be quantified using the EI ratio and correlation (see Methods), which, respectively, measure how equal and coupled excitatory and inhibitory currents are over time (Vogels et al., 2011; Taylor et al., 2024b).

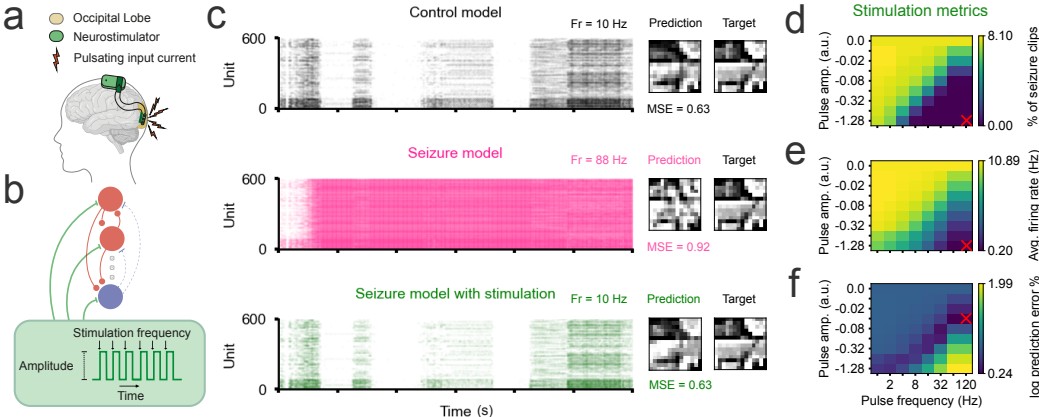

Figure 6: **Preventing seizure-like activity with pulsating input currents. a.** Adapted schematic from (Baud et al., 2018) showing an RNS system delivering current to the seizure onset zone in the occipital lobe. **b.** The schematic illustrates the current injection into each unit in the seizure model, with the stimulation parameters shown in the green box. **c.** Raster plots comparing neural responses in the control model (black), unstimulated seizure model (pink), and seizure model with stimulation (green). MSE values quantify the difference between predicted and target frames. **d.** Heatmap of seizure prevalence as a function of stimulation frequency (x-axis) and current amplitude (y-axis). **e.** Corresponding heatmap of average firing rates (Hz). **f.** Corresponding heatmap of the log-scaled difference in prediction error between the stimulated seizure and control model. Red crosses mark the optimal frequency and amplitude for each measurement.

The seizure and control models had similar EI ratios (median $\sim 0.6 - 0.7$) over the non-provocative stimulus input (Fig. 5d), resembling the EI ratios of V1 neurons in mice (Xue et al., 2014; Banerjee et al., 2016; Adesnik, 2018). The seizure model's EI ratio significantly increased over the provocative stimulus inputs (median $6.85$). In contrast, the control model's EI ratio increased only slightly (median $1.23$). Similarly, we found the EI currents to be notably correlated over time across the units in the seizure (median $0.65$) and control (median $0.72$) models to the non-provocative stimulus input (Fig. 5e). The EI correlation significantly dropped in the seizure model over the provocative stimulus input (median $0.39$), while the control model's EI correlation remained largely unchanged (median $0.73$).

### 3.5 Minimizing seizure occurrence through pulsating input currents

One treatment option for patients unresponsive to anti-epileptic drugs are neuronal stimulation techniques such as RNS (Baud et al., 2018). Here, a multielectrode array is implanted in close proximity to the seizure onset zone (Fig. 6a) (Baud et al., 2018). The electrode emits a pulsating stimulatory current to prevent or terminate seizure-like activity. We explored if the seizure model's seizure-like activity could be reduced by inputting pulsating input current into the units (Fig. 6b).

Surprisingly, we were able to prevent seizure-like activity in response to the provocative seizure-inducing *Pokemon* segment (Hermes et al., 2017) by inputting particular stimulatory currents (Fig. 6c). The spike raster of the stimulated seizure model resembled that of the control model with a firing rate of $\sim 10$Hz, significantly lower than the unstimulated seizure model's firing rate of $88$Hz. Additionally, the stimulated seizure model's predicted future frames qualitatively matched those of the control model and did not contain the noisy artifacts seen in the unstimulated model. The prediction accuracy of the stimulated seizure model was similar to the control model ($\sim 0.63$ MSE), marking a significant improvement over the unstimulated seizure model ($\sim 0.92$ MSE).

We explored the space of stimulation frequencies (logarithmically spaced between $0$ and $120$Hz) and amplitudes (logarithmically spaced between $0$ and $-1.28$ a.u.) to determine the optimal values for reducing seizure-like activity in response to the natural stimulus dataset. The percentage of clips triggering seizure-like activity declined as a function of increasing stimulation amplitude and frequency (Fig. 6d). For the largest tested frequency and amplitude values, we found seizure-like

activity completely disappeared. However, this is undesirable as activity was completely silenced (Fig. 6e).

As an additional evaluation metric, we compared the future frame prediction accuracy of the stimulated seizure model to that of the control model (Fig. 6f). The future-frame prediction accuracies became more similar as a function of increasing stimulation amplitude and frequency. However, the prediction accuracies diverged when the stimulation parameters became too large, due to the units becoming more quiescent. We found the prediction accuracies to be most similar (1.7% difference) when stimulating with an amplitude of $-0.02$ a.u at a frequency of $\sim 64 - 120$Hz. The parameters reduced seizure-like activity occurrence from 7.8% to 1.5%, indicating their effectiveness in restoring baseline model activity and function.

## 4   DISCUSSION

**Mechanism**   We investigated the theory that PE results from abnormal GABAergic function by weakening inhibitory synapses in a spiking network model of V1. This led to abnormally elevated firing rates, particularly in response to high-contrast stimuli (Fig. 4b), with persistent activity even after stimulus removal (Fig. 5a). Thus, our findings support the notion of inhibition stabilizing network activity and preventing runaway excitation (Sadeh & Clopath, 2021; Adesnik, 2017). Notably, while the gain response is higher in the seizure model, tuning characteristics remain similar to the control model (Fig. 3), reflecting findings in biology (Atallah et al., 2012; Katzner et al., 2011). This may explain how altered excitatory and inhibitory dynamics in PE patients result in seizures, but visual processing remains normal during interictal periods.

**Translation**   High-frequency current stimulation in the seizure model restored baseline firing activity and function as measured by the model's prediction MSE (Fig. 6c). These results align with high-frequency RNS treatment in patients which effectively reduces seizures (Alcala-Zermeno et al., 2023). High-frequency stimulation is proposed to inhibit targeted structures, yet its exact mechanisms remain unknown (Bikson et al., 2001; Foutz & Wong, 2022). Interestingly, the most effective neural stimulation frequency range (64-120Hz) used in our model overlaps with gamma oscillations (30-70Hz) in human V1, which are implicated in seizures resulting from high-contrast gratings (Hermes et al., 2017). Thus, high-frequency stimulation may be disrupting these synchronous gamma oscillations (Sadeh & Clopath, 2021) and preventing seizure onset. Our seizure model also extends to computational theories of cortical function and dysfunction. Predictive processing is hypothesized to be fundamental to healthy V1 functioning (Keller & Mrsic-Flogel, 2018). We computationally showed inhibitory synapses to be crucial for maintaining healthy predictive function in the spiking model, where seizure-like activity impaired the model's predictive performance (Fig. 6c). These findings may extend to biological theories linking prediction errors to pathology (Horga et al., 2014; Van de Cruys et al., 2014).

**Limitations**   Defining seizure-like activity in our model is challenging due to the broad definition of seizures in humans (Truccolo et al., 2011) and the lack of a validated seizure-inducing dataset. We therefore validated our model using various clips reported to induce seizures. Another limitation is the validation of results against animal studies (mice and monkeys). Animal models have been extremely valuable to understand epilepsy dynamics (Grone & Baraban, 2015). However, it is unknown to what extent the findings we compare to animal models (Fig. 3a and Fig. 4d, e) translate to human patients.

**Impact and Ethics**   Adolescents, who are already at a higher risk of developing PE (Fisher et al., 2005), face increased seizure risks with the surge of video consumption on platforms like TikTok. As testing harmful content on patients with PE is unethical, our model could screen social media content for provocative stimuli. Current warning systems screen for features such as 'high-contrast stimuli' and 'pulsating light stimulation' (Fisher et al., 2022), and therefore may miss certain stimuli or overly restrict content. Our model is not trained to identify particular stimuli and may be better suited to identify harmful features that do not fall into these categories. However, it is important to note that while our model offers a framework to test the underlying pathophysiology of PE, it does not fully replicate the complexity of human physiology.

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

## 5 APPENDIX / SUPPLEMENTAL MATERIAL

### 5.1 SPIKING TEMPORAL PREDICTION MODEL

The spiking temporal prediction model was implemented as a population of $N = 600$ recurrently-connected excitatory and inhibitory spiking units, defined by the following equations (Taylor et al., 2024b):

$$I_i[t] = b_i^{\text{in}} + \underbrace{\sum_{t'=1}^{T_E} \sum_{h=1}^{H} \sum_{w=1}^{W} W_{it'hw}^{\text{in}} x_{hw}[t - t' + 1]}_{\text{Feedforward current}} + \underbrace{\sum_{j=1, j \neq i}^{N} W_{ij}^{\text{rec}} S_j[t-1]}_{\text{Recurrent current}} \tag{1}$$

$$V_i[t] = \left( \beta_i V_i[t-1] + (1 - \beta_i) I_i[t] \right) \left( 1 - S_i[t] \right) \tag{2}$$

$$S_i[t] = \begin{cases} 1 & \text{if } V_i[t] > 1 \\ 0 & \text{otherwise} \end{cases} \tag{3}$$

$$\hat{y}_{hw}[t] = b^{\text{out}} + \sum_{i=1}^{N} \sum_{t'=1}^{T_D} W_{it'hw}^{\text{out}} S_i[t - t' + 1] \tag{4}$$

$$W_{ij}^{\text{rec}} = \begin{cases} -\alpha |\tilde{W}_{ij}^{\text{rec}}| & \text{if } j < N_I \\ |\tilde{W}_{ij}^{\text{rec}}| & \text{otherwise} \end{cases} \tag{5}$$

**Input current** The input current $I_i[t] \in \mathbb{R}$ (Eq. 1) to unit $i$ at time step $t$ is calculated from the input stimulus $x \in \mathbb{R}^{T \times H \times W}$ (of $T = 42$ frames, and spatial height $H = 20$px and spatial width $W = 20$px); model output spikes $S[t-1] \in \mathbb{R}^N$ from the previous time step; and a bias term $b_i^{\text{in}} \in \mathbb{R}$. The feedforward connectivity $W^{\text{in}} \in \mathbb{R}^{N \times T_E \times H \times W}$ (of $T_E = 15$ temporal frame span) maps the input stimulus into a feedforward current contribution and the recurrent connectivity $W^{\text{rec}} \in \mathbb{R}^{N \times N}$ maps previous output spikes into a recurrent current contribution. The first 5 spatial frames of the feedforward connectivity were zero-masked to emulate the transduction and transmission latency to V1 (Land et al., 2013; Kirchberger et al., 2023) and Gaussian noise $\epsilon_p \sim \mathcal{N}(0, 0.2^2)$ was added to the input stimulus to emulate the noisy photoreceptors in the retina (Baylor et al., 1980). 15% of the model unit population were assigned to be inhibitory ($N_I = 90$ inhibitory units and $N_E = 510$ excitatory units) to mimic the biology (Sillito, 1975; Fitzpatrick et al., 1987; Hofer et al., 2011). All outward connection weights from unit $j$ are either positive (*i.e.* excitatory) or negative (*i.e.* inhibitory), which were enforced by taking the absolute value of every weight in the unconstrained recurrent weight matrix $\tilde{W}^{\text{rec}}$, and appropriately negating connection weights from unit $j$ to $i$ (Eq. 5). In this work, we introduced parameter $\alpha$ which we used to weaken the inhibitory connections in the model by multiplying their value by $\alpha$ (where $\alpha = 1$ is a 100% weight decrease to zero).

**Spiking dynamics and predicted output** All spiking units implement the normalized and discretized leaky integrate-and-fire (LIF) model (Taylor et al., 2024a), which decays the membrane potential $V_i[t] \in \mathbb{R}$ of unit $i$ at time step $t$ by learnt factor $\beta_i$ (Eq. 2). The membrane was also multiplicatively perturbed using Gaussian sampled noise $\epsilon_g \sim \mathcal{N}(0, 0.6^2)$ to emulate the synaptic noise of V1 neurons (Rusakov et al., 2020). In the event of a spike occurring in the preceding time step (which happens when the membrane potential reaches the firing threshold equal to one), the membrane potential resets to zero (Eq. 3). A prediction of the future spatial movie frame $\hat{y}[t] \in \mathbb{R}^{H \times W}$ was generated (with $\hat{y}_{hw}[t] \in \mathbb{R}$ denoting the predicted pixel value) at every time step $t$, by mapping a span of $T_D = 2$ temporal frames of proceeding spike activity using weights $W^{\text{out}} \in \mathbb{R}^{N \times T_D \times H \times W}$, plus a bias $b^{\text{out}} \in \mathbb{R}$ (Eq. 4).

**Loss function and training** The original model was trained by minimizing the loss

$$\mathcal{L}_{\text{total}} = \mathcal{L}_{\text{prediction}} + \lambda \mathcal{L}_{\text{metabolic}} \tag{6}$$

where $\mathcal{L}_{\text{normative}}$ is the prediction loss (how well the model predicts future movie frames) and $\mathcal{L}_{\text{metabolic}}$ is the metabolic loss (a biological regularization that jointly penalizes connectivity and

activity, see (Taylor et al., 2024b) for more details) weighted by hyperparameter $\lambda$. All training was performed using a NVIDIA GeForce RTX 3090 GPU which took approximately $\sim 30$ hours. PyTorch (Paszke et al., 2019) and DevTorch (Taylor, 2024) were used for model implementation and training and BrainBox was used for analyzing the spike statistics, receptive fields and tuning properties. See (Taylor et al., 2024b) for more details. We used the pre-trained model in this work and modified the inhibitory weights post training.

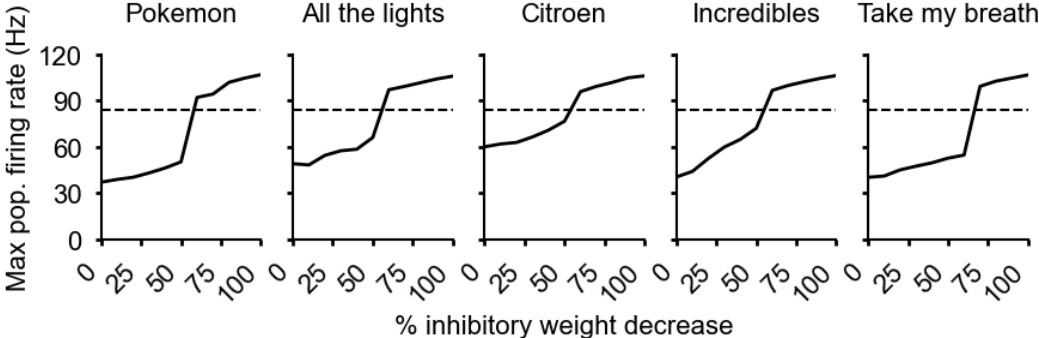

Figure 1: **Seizure model's firing rate to provocative stimuli.** Every column plots the maximum population firing rate (Hz) (y-axis) over the duration of the stimulus for the V1 model with different inhibitory weight % decreases (x-axis). $0\%$ decrease corresponds to the control model and $100\%$ corresponds to the seizure model with all inhibitory connections completely ablated. Each stimulus clip (*Pokemon, All of the Lights, Citroen, Incredibles 2, Take My Breath*) has been reported to induce seizures in patients with PE (Hermes et al., 2017; Sweney, 2012; Svachula, 2018; Aswad, 2021). The dotted vertical line marks the firing rate threshold we used to identify model activity as seizure-like and stimulus as provocative.

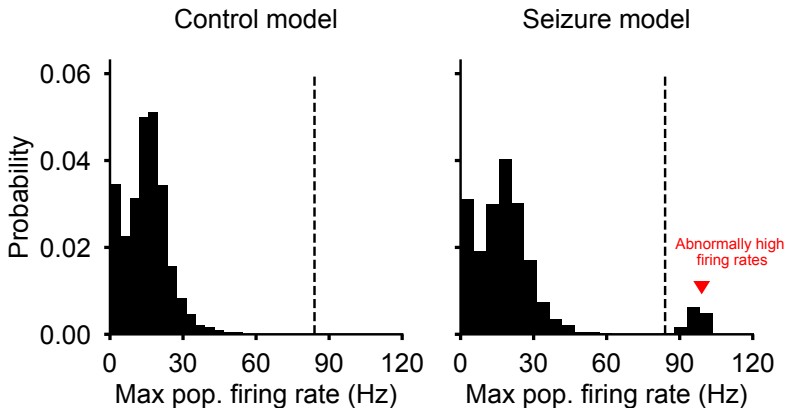

Figure 2: **Maximum population firing rate distribution over the natural stimulus dataset.** The dotted vertical line marks the firing rate threshold we used to classify model activity as seizure-like and stimulus as provocative. For the control model (left) we identified no seizure-like activity. For the seizure model (right) we identified a few clips with seizure-like activity (which the chosen threshold captures).

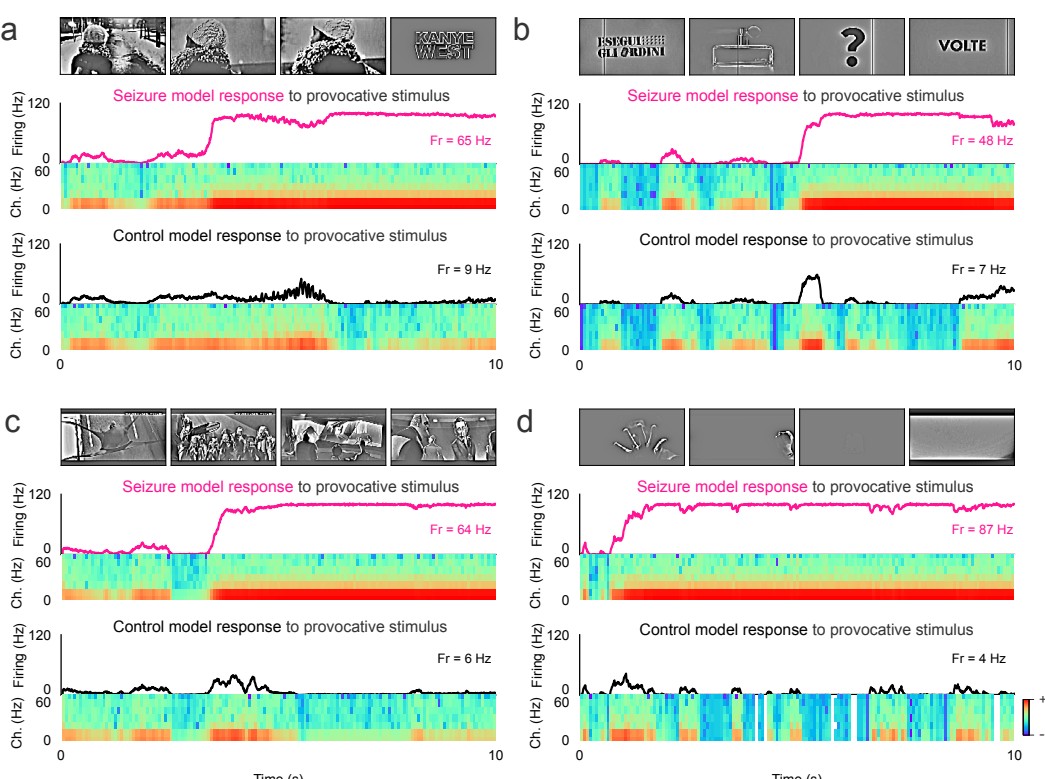

Figure 3: **Capturing seizure-like activity to additional provocative stimuli.** Mean population firing activity of the seizure model (pink) and control model (black) in response to the **a.** *All of the Lights* music video, **b.** *Citroen* advert, **c.** *Incredibles 2* movie and **d.** *Take my breath* music video. Online reports suggest these stimuli to induce seizures in patients with PE (Concepcion, 2013; Sweney, 2012; Svachula, 2018; Aswad, 2021). Spectrograms plot the frequency-time power representation of the model activity, with the highest and lowest power denoted in red and blue respectively.

| metric | Unit type | Control model | Seizure model | p-value |
|---|---|---|---|---|
| OSI | Excitatory | $0.82 \pm 0.35$ | $0.81 \pm 0.35$ | 0.4709 (NS) |
|  | Inhibitory | $0.09 \pm 0.19$ | $0.13 \pm 0.19$ | 0.0010 (***) |
| DSI | Excitatory | $0.12 \pm 0.13$ | $0.15 \pm 0.14$ | 0.0008 (***) |
|  | Inhibitory | $0.05 \pm 0.04$ | $0.10 \pm 0.10$ | 0.0002 (***) |

Table 1: **Tuning statistics between the control and seizure model.** Statistical significance assessed using the Mann-Whitney U test (***, **, and * denote p ¡ 0.001, p ¡ 0.01, and p ¡ 0.05, respectively).

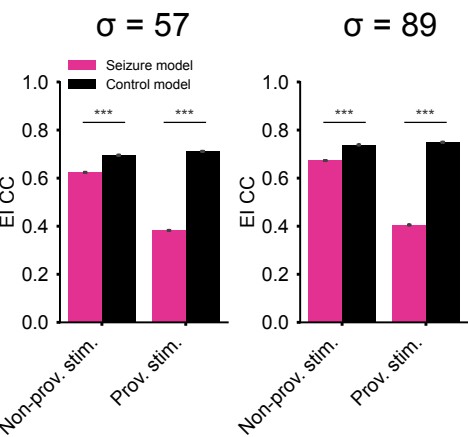

Figure 4: **EI correlation coefficients calculated using Gaussian spike-train smoothing kernel with different $\sigma$.** Bars plot the median and bootstrapped standard error, with statistical significance assessed using the Mann-Whitney U test (***p< 0.001).

