# OpenReview forum: "Spike-to-excite: photosensitive seizures in biologically-realistic spiking neural networks"
_ICLR.cc/2025/Conference — Submitted to ICLR 2025_

### Official Review · Reviewer_shUi · 2024-10-30

**Soundness:** 2
**Presentation:** 3
**Contribution:** 2
**Rating:** 3
**Confidence:** 4

**Summary:**

The paper addresses seizure-like activity emerging in a recurrent E-I spiking network trained to efficiently represent natural scenes. Authors show that if an efficient network has weak inhibition, it enters seizure-like activity in response to provocative stimuli. The model has been developed previously, but the paper addresses a partially unclear mechanism of seizure generation and propagation in biological brains in response to provocative stimuli.

**Strengths:**

The paper is well written - very clear and mostly transparent about what is done and what is achieved. The paper also motivates convincingly why the main question is of interest. The paper is technically sound, even though it lacks transparency. It presents minor novel findings.

**Weaknesses:**

Weaknesses:
1) The main weakness is lack of significance of the main finding. Observing strong firing rates in a spiking network with strong input and too weak inhibition is a highly expected result. Also, observing weaker correlation between E and I currents in a network with too weak inhibition seems close to trivial.

2) Another weakness is the that the model is not well described. Authors use a previously published model. While the previous work that developed the model is cited, authors should nevertheless describe it in some detail. In particular, model properties that are relevant for the seizure-like activity should be made clear to the reader, otherwise it is very difficult to appreciate the results.

3) Authors do not provide (at least not in the main text where it would belong) important parameters of the model, such as the network size, the ratio of E-I neuron numbers, the type of connectivity (random / structured, sparse / dense, symmetric / asymmetric), and the mean connectivity strength. I suspect that used parameters are important for seizure-like activity. Even if they are not, this should be discussed in the paper.

4) Authors mention that they convolve currents with a Gaussian kernel with standard deviation of 72 ms. This seems to be a very long time constant that potentially obscures finer fluctuations of currents. Why this choice?

**Questions:**

1) My main question to the Authors is if the model can go into seizure-like activity also without a provocative stimulus? Achieving strong firing rates after weakening the inhibition seems like a trivial result, as it is highly expected. Are there parameter settings where seizures appear without a provocative stimulus? If so, how are these settings different from settings where seizures only appear with a provocative stimulus?

2)   Besides the change in the correlation of EI currents, is there another statistics of the activity in response to non-provocative stimuli (or during spontaneous activity) that differ between the seizure model and the control model?

3) How does training the model on efficient prediction influence the results? Is the model that is less efficient more likely to display seizures?

4) The feedforward input to the network is a current generated from movie clips. A neural signal incoming to the primary visual cortex in a biological brain has been processed and changed by the retina and the subcortical processing. While I find it acceptable to simplify these processing steps by simply converting the movie clips into an input current, the authors should comment on that in the text, possibly as a limitation of the model.

5) Authors report that, in order to achieve seizure-like activity, they weaken the inhibition, but it would be useful to know the numbers. What si the inhibition strength in the efficient network and in the network with seizures?

---

### Official Review · Reviewer_szyQ · 2024-10-31

**Soundness:** 2
**Presentation:** 2
**Contribution:** 2
**Rating:** 3
**Confidence:** 5

**Summary:**

This paper presents a biologically realistic spiking neural network (SNN) model of the primary visual cortex (V1) to study photosensitive epilepsy (PE), specifically examining seizure-like activity triggered by visual stimuli. The authors simulate weakened inhibitory connections in the model to replicate the neural conditions associated with PE and observe seizure dynamics in response to high-contrast and flashing visual inputs. The study finds that reducing inhibitory strength in V1 increases the model’s sensitivity to seizure-inducing stimuli, and that applying responsive neurostimulation-like (RNS) pulse currents can mitigate these effects. The contributions include the first computational model of PE with V1-like properties, new insights into E/I imbalance in seizure conditions, and potential implications for clinical treatments and real-time content screening for seizure triggers.

**Strengths:**

1 The paper introduces a  spiking neural network model to simulate photosensitive epilepsy (PE), targeting a gap in epilepsy modeling by focusing on seizure induction from visual stimuli.
2 The model is methodologically rigorous, using biologically realistic features and controlled testing with seizure-inducing stimuli, which strengthens the study's credibility.
3 This work offers a new tool for studying visual cortex-related seizure disorders and has clinical relevance.

**Weaknesses:**

1 The paper’s core contribution relies on a widely assumed mechanism—the E/I (Excitation-Inhibition) imbalance that leads to seizure-like activity. This hypothesis has been extensively validated in the literature (e.g., Sadeh & Clopath, 2021; Shao et al., 2019), yet the authors do not provide any novel insights or perspectives on it.

2 Although the model attempts to approximate a biologically realistic spiking neural network (SNN) architecture, relying solely on weakened inhibitory synapses to simulate PE lacks the complexity of actual epilepsy, which often involves a variety of neurotransmitters and molecular channels.

3 The validation relies on simplistic metrics, primarily firing rates and E/I balance under different stimuli. The study misses other crucial electrophysiological characteristics, such as neuron synchrony or spectral features.

4 While the paper introduces pulse stimulation to mimic RNS therapy for suppressing seizure activity, the tested parameter space and stimulation methods are simplistic, lacking consideration for how different frequencies might affect E/I balance long-term.

5 The authors test the model with known seizure-inducing video clips but do not clarify whether these stimuli elicit responses consistent with those of actual PE patients.

**Questions:**

1 Can you provide a more detailed justification for relying solely on E/I imbalance as the central mechanism for photosensitive epilepsy (PE)? Specifically, how does this approach offer novel insights beyond the well-established role of E/I imbalance in seizure activity?
2 Your model assumes that weakening inhibitory synapses alone can simulate PE. Could you discuss any limitations in this approach, given the known complexity of epilepsy involving multiple neurotransmitters and cellular pathways?
3 The current model uses fixed parameters for pulse stimulation, similar to responsive neurostimulation (RNS). Would varying the stimulation frequency, amplitude, or electrode location impact the model’s outcomes? Have you tested alternative configurations?
4 What are the limitations of this model in replicating the complexity of human brain dynamics in epilepsy?

**Details Of Ethics Concerns:**

no concerns

---

### Official Review · Reviewer_MDRx · 2024-11-04

**Soundness:** 2
**Presentation:** 3
**Contribution:** 2
**Rating:** 5
**Confidence:** 3

**Summary:**

The authors utilize a  spiking neural network model, trained on predictive loss,  and show that decreasing inhibitory connections causes seizure-like behavior. By manipulating the stimulus and investigating input currents, the authors demonstrate that seizures are self-sustaining due to an imbalance of E/I currents.

**Strengths:**

The study is a great example of machine-learning trained models of neural systems, which can then be used to investigate neuroscience principles. The investigation of seizure-model receptive fields, along with masked visual inputs, back up the E/I current investigation of 3.4 to further back up the hypothesis that the model parameters are responsible for sustained seizure activity.

**Weaknesses:**

Ultimately the manipulation of the neural model is fairly brief, with the majority of the manuscript describing datasets or model behavior. This main raise concerns about interest to the broader ICLR community.

**Questions:**

-	Are there other properties of the model that could be investigated to delve deeper into the general ML applicability? Perhaps altering the membrane time constants in addition to overall weight balance, to demonstrate that weight-by-dynamics interactions affect network stability. Or somehow altering the tight versus loose EI balance to investigate how tightly coupled the balance must be to maintain healthy behavior. These are just suggestions, but I do think some additional investigation to the neural model is required to warrant acceptance to a machine learning conference.
-	The training method in Appendix 5.1 are insufficiently detailed to replicate the study, without referencing the originating paper. What is the surrogate training function, optimizer, etc.

---

### Meta-Review · Area_Chair_4kqE · 2024-12-21

**Metareview:**

This paper describes a spiking neural network model designed to capture seizure-like neural activity found in photosensitive epilepsy.  The reviewers agreed that this is an important and interesting problem, and that the study is methodologically rigorous and technically sound. Unfortunately, however, they raised substantial concerns about clarity and completeness of the manuscript, as well as about the significance of the findings. I regret that the paper in its current form cannot be accepted to this year's ICLR, but the work seems promising and I wish the authors the best of luck in revising it for publication elsewhere.

**Additional Comments On Reviewer Discussion:**

The authors did not write a rebuttal, so there was no discussion during the rebuttal period.

---

### Decision · Program_Chairs · 2025-01-22

Reject